# RoHOI: Robustness Benchmark for Human-Object Interaction Detection

## Abstract

Human-Object Interaction (HOI) detection is crucial for robot-human assistance, enabling context-aware support. However, models trained on clean datasets degrade in real-world conditions due to unforeseen corruptions, leading to inaccurate prediction. To address this, we introduce the first robustness benchmark for HOI detection, evaluating model resilience under diverse challenges. Despite advances, current models struggle with environmental variability, occlusions, and noise. Our benchmark, RoHOI, includes 20 corruption types based on the HICO-DET and V-COCO datasets and a new robustness-focused metric. We systematically analyze existing models in the HOI field, revealing significant performance drops under corruptions. To improve robustness, we propose a Semantic-Aware Masking-based Progressive Learning (SAMPL) strategy to guide the model to be optimized based on holistic and partial cues, thus dynamically adjusting the model's optimization to enhance robust feature learning. Extensive experiments show that our approach outperforms state-of-the-art methods, setting a new standard for robust HOI detection. Benchmarks, datasets, and code will be made publicly available.

## 1 Introduction

Human-Object Interaction (HOI) detection aims to identify and interpret interactions between humans and objects Kim et al. (2021); Hou et al. (2021a); Zhang et al. (2022b); Tamura et al. (2021); Kim et al. (2022); Hou et al. (2021b). It serves many critical applications, including autonomous driving Martin et al. (2019); Peng et al. (2022), robotics Xu et al. (2020), video surveillance Dogariu et al. (2020); Prest et al. (2013), and augmented reality Zhang et al. (2025), where a fine-grained understanding of human behavior and object handling is paramount Gupta & Malik (2015).

In recent years, transformer-based models Zhang et al. (2022b); Tu et al. (2023); Zou et al. (2021) and Vision-Language Models (VLMs) Cao et al. (2023); Gao et al. (2024); Chen et al. (2024b), have achieved impressive success in the HOI detection field. These models excel at capturing intricate relationships between human-object pairs by leveraging large-scale datasets and contextual information.

Despite these advancements, existing models remain sensitive to real-life corruptions, particularly semantic distortions like occlusions and perspective shifts, as well as environmental variations. Even minor disturbances may pose significant risks in critical applications like autonomous driving and robotics. Enhancing robustness under these conditions remains a crucial but unresolved challenge. Existing HOI detection methods Zhang et al. (2022b); Tu et al. (2023); Cao et al. (2023) had their performance evaluated predominantly under ideal conditions. Thus, a dedicated robustness benchmark tailored explicitly to HOI detection, covering both pixel-level and semantic-level corruptions, is urgently required to put their resilience to the test.

In this paper, we introduce the **Ro**bustness benchmark for **HOI** detection (**RoHOI**), systematically assessing HOI detection models under 20 diverse forms of data corruption. These were designed to simulate real-world conditions. The HICO-DET Chao et al. (2018) and V-COCO Gupta & Malik (2015) datasets served as starting point. We include robustness-focused metrics, such as the Mean Robustness Index (MRI) Hendrycks & Dietterich (2019) and our novel Composite Robustness Index (CRI). Through our evaluations of existing HOI detection methods on the newly proposed robustness

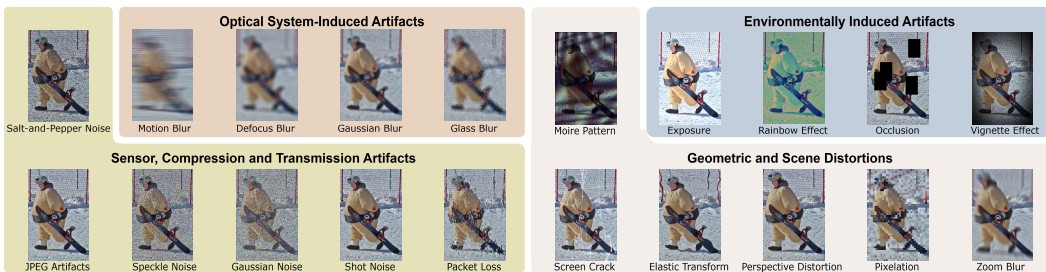

Figure 1: Our RoHOI dataset comprises 20 types of algorithmically generated corruptions, systematically categorized into four groups. Each corruption type has five levels of severity, leading to a total of 100 distinct corruptions. These corruptions simulate realistic semantic and structural disturbances specifically encountered in practical application.

test set, we observe that their performance is less than optimal. This finding underscores the need to develop more robust HOI detection models.

To mitigate the issue of performance degradation, we propose our Semantic-Aware Masking-based Progressive Learning (SAMPL) method. Specifically, we utilize SAM Kirillov et al. (2023) to identify HOI-relevant regions and subsequently apply a dynamic masking strategy with varying size ratios. This approach enables the model to focus on both holistic and partial information, thereby enhancing its robustness throughout the optimization process.

Our proposed learning strategy delivers superior HOI detection, retaining state-of-the-art (SOTA) performance on a clean test set while delivering substantial improvements under corruptions. This highlights the model's enhanced resilience and reliability in challenging real-world conditions.

The main contributions of this paper are summarized as follows:

- We for the first time introduce a robustness benchmark specifically tailored to HOI detection. Our benchmark incorporates 20 distinct corruptions that closely mirror real-world challenges and a set of robustness-focused metrics.

- We propose a novel semantic-aware masking curriculum learning approach that adaptively guides the model to optimize learning on both holistic and partial information.

- SAMPL yields SOTA in terms of robustness, surpassing prior HOI models under both clean and corrupted conditions.

## 2    RELATED WORK

**Human-Object Interaction Detection.** Human-object interaction (HOI) detection Yao & Fei-Fei (2010); Fang et al. (2018); Gupta et al. (2019); Li et al. (2019); Wan et al. (2019); Park et al. (2018); Xu et al. (2020); Yu et al. (2018); Zhang et al. (2015); Zhou et al. (2020) aims to automatically identify and interpret relationships between humans and objects, typically divided into two categories: *two-stage* and *one-stage* approaches.

*Two-stage HOI detection methods* Qi et al. (2018); Xu et al. (2019); Liang et al. (2021); Zhou & Chi (2019) first detect humans and objects independently before classifying interactions. Early CNN-based approaches Xu et al. (2019); Zhou & Chi (2019); Qi et al. (2018) mainly utilized spatial contexts for interaction modeling. Later graph-based methods Zhou & Chi (2019); Xu et al. (2019); Ulutan et al. (2020); Wang et al. (2021); Zhang et al. (2021b) improved relational reasoning explicitly. Subsequent methods further enhanced precision via pose-aware representations Wan et al. (2019), transformer-based unpaired modeling Zhang et al. (2022a), and integration-decomposition strategies Li et al. (2020).

*One-stage HOI detection methods* Yuan et al. (2023); Zhang et al. (2023) predict human–object interactions in a single forward pass, improving efficiency. Early approaches rely on point-level matching (e.g. PPDM Liao et al. (2020)) or union-level ROI features (e.g. UnionDet Kim et al.

(2020)). Later methods refine localization via glance-and-gaze attention (e.g. Zhong et al. (2021)). Transformer-based pipelines further enrich relational context, and methods like RLIPv2 Yuan et al. (2023) combine vision and language modalities for enhanced semantic generalization.

Recent trends explore generative priors and robust-aware training. DIFFUSIONHOI Li et al. (2024) integrates a text-to-image diffusion model to derive relation-aware prompts, while HOI-IDiff Hui et al. (2025) recasts each human–object prediction as an "HOI image" and generates it with a customized diffusion process. UAHOI Chen et al. (2024a) explicitly models prediction uncertainty and incorporates it into the objective, yielding more stable interaction predictions. Although promising, these approaches are still evaluated on clean benchmarks. In contrast, our work introduces RoHOI, a dedicated benchmark of 20 corruption types, and proposes SAMPL to strengthen robustness under perturbation.

**Visual Robustness Benchmarks.** Robustness benchmarks have been widely established to assess visual model resilience across various tasks Chattopadhyay et al. (2021); Wang et al. (2024); Sarkar et al. (2024); Schiappa et al. (2024); An et al. (2024); Liu et al. (2024b); Yi et al. (2024); Kamann & Rother (2020); Ma et al. (2024b); Hao et al. (2024); Xie et al. (2025); Hao et al. (2025); Liu et al. (2024c); Michaelis et al. (2019); Zeng et al. (2024), including image classification (ARES-Bench Liu et al. (2024a)), visual question answering Ishmam et al. (2024), common corruption evaluation (ImageNet-C Hendrycks & Dietterich (2019)), attribute editing robustness (ImageNet-E Li et al. (2023)), one-shot skeleton-based action recognition under occlusion Peng et al. (2023), and robust document layout analysis (RoDLA Chen et al. (2024c)). However, robustness benchmarks explicitly designed for HOI detection have not been explored. In this work, we address the uncharted robust HOI detection area by constructing the new RoHOI benchmark and by putting forward the SAMPL method to achieve more reliable HOI detection performance.

## 3 BENCHMARK

**Corruptions.** To rigorously assess the robustness of HOI detection, we introduce our dedicated RoHOI benchmark, designed to evaluate model performance under real-world corruptions. Existing benchmarks such as ImageNet-C Hendrycks & Dietterich (2019) primarily cover low-level pixel disturbances like brightness or blur. In contrast, RoHOI explicitly includes semantic-level corruptions specifically tailored to challenge spatial and relational reasoning critical for HOI detection. RoHOI explicitly introduces 20 corruptions derived from HICO-DET and V-COCO images, categorized into four groups: optical system artifacts such as motion blur and glass blur; sensor, compression, and transmission distortions such as packet loss and JPEG artifacts; environmental effects including exposure imbalance and occlusion; and geometric or structural disruptions such as screen cracks, elastic deformation, and perspective distortion. The corruptions range from pixel-level noise to semantic-level challenges that affect spatial alignment and interaction reasoning. Each corruption is implemented with five severity levels to support detailed analysis under progressively more difficult conditions.

*Optical System-induced (OS) Artifacts* degrade spatial clarity and obscure interaction details by distorting object boundaries. Our benchmark includes *motion blur (MB)*, *defocus blur (DB)*, and *Gaussian blur (GauB)*, which simulate degradation caused by camera movement, defocus, and signal processing. We refine *glass blur (GB)* to better replicate distortions caused by imaging through uneven or scratched glass surfaces.

*Sensor, Compression and Transmission (SCT) Artifacts* stem from hardware limitations, lossy encoding, and data transmission errors. Alongside standard corruptions like *Gaussian noise (GauN)*, *shot noise (ShN)*, *salt-and-pepper noise (S&P)*, *JPEG artifacts (JPEG)*, and *speckle noise (SN)*, we introduce *packet loss (PL)*, which simulates data degradation due to incomplete transmissions.

*Environmentally Induced (EI) Artifacts* alter image quality due to external conditions. Extreme lighting variations cause *exposure (EXP)* distortion. The *rainbow effect (RE)* misaligns color channels, distorting object hues. *Occlusion (OCC)* removes critical visual information when foreign objects block key scene elements, while the *vignette effect (VE)* creates uneven brightness, reducing visibility at image borders.

*Geometric and Scene (G&S) Distortions* alter spatial relationships and object proportions. *Moire patterns (MP)* create artificial wave-like distortions in high-frequency textures. *Screen crack (SC)*

introduce irregular fractures and glare artifacts, while *elastic transformation (ET)* warp object structures. *Perspective distortion (PD)* skews object shapes when captured at extreme angles. *Pixelation (PIX)* replaces fine details with block-like artifacts due to low-resolution rendering, and *zoom blur (ZB)* degrades object sharpness through rapid motion along the optical axis.

**Datasets.** To evaluate robustness under real-world corruptions, we use two widely adopted HOI detection datasets: HICO-DET and V-COCO. HICO-DET Wang et al. (2020) contains $47,776$ images with 600 interaction classes derived from 117 action classes and 80 object categories. Each image is annotated with human-object pairs and their corresponding interactions, making it a benchmark for evaluating diverse HOI scenarios. V-COCO Gupta & Malik (2015) is a subset of MS-COCO Lin et al. (2014) with $10,346$ images, $16,199$ human instances, 81 object categories and 29 verb classes. Unlike HICO-DET, which provides a fine-grained benchmark, V-COCO evaluates model performance on common human-object interactions. Evaluating these datasets under corruption provides a systematic robustness assessment beyond clean-image performance.

**Baselines.** To systematically assess robustness, we benchmark representative two-stage and one-stage HOI detectors spanning transformer pipelines and vision–language training paradigms. For the two-stage family, we include UPT Zhang et al. (2022a), which detects humans and objects before interaction classification using a unary–pairwise transformer, and GEN-VLKT Liao et al. (2022), which adopts a guided-embedding association design and a visual–linguistic knowledge transfer (VLKT) strategy that transfers CLIP text embeddings via initialization and a mimic loss rather than performing generic VLM pretraining. For the one-stage family, we evaluate QPIC Tamura et al. (2021) as a query-based pairwise transformer detector, CDN Zhang et al. (2021a) as an end-to-end cascade disentangling network, and DiffHOI Yang et al. (2023) which injects text-to-image diffusion priors to enrich relational cues. We further include FGAHOI Ma et al. (2024a) with fine-grained anchors, QAHOI Chen & Yanai (2023) with query-based anchors, MUREN Kim et al. (2023) with multiplex relational context learning, and SOV-STG Chen et al. (2023) with subject–object–verb decoding guided by specific-target denoising. Finally, we consider RLIPv2 Yuan et al. (2023),which incorporates region-language alignment for fine-grained interaction reasoning. Additionally, we develop SAMPL, which is built upon RLIPv2 and incorporates semantic-aware masking-based progressive learning. Our benchmark systematically evaluates these models across diverse corruption types and severity levels, providing a comprehensive assessment of HOI detection robustness.

**Evaluation Metrics.** We evaluate the clean test set of HICO-DET Wang et al. (2020) using mAP at Intersection over Union (IoU) 0.5 for all (Full), rare ($<10$ instances), and non-rare ($\geq 10$ instances) categories Yuan et al. (2023). For the clean test set of V-COCO Gupta & Malik (2015), we use $\text{AP}_{\textbf{role}}$ for interaction detection, with $\text{AP}_{\textbf{role}}^{\#1}$ for action recognition and $\text{AP}_{\textbf{role}}^{\#2}$ requiring object localization at IoU 0.5. To evaluate HOI detection performance on our robustness benchmark, we introduce two robustness-focused metrics: Mean Robustness Index (MRI) and Composite Robustness Index (CRI). MRI, following the definition in prior work Hendrycks & Dietterich (2019), measures average performance under corruptions but overlooks stability across severity levels. To address this, we propose CRI, which penalizes performance variance while normalizing against clean-set accuracy, ensuring models with consistent robustness are appropriately rewarded.

*Mean Robustness Index (MRI)* serves as a baseline measure of a model's average performance across corruption types and severities:

$$\text{MRI} = \frac{1}{C} \sum_{c=1}^{C} \left( \frac{1}{L_c} \sum_{l=1}^{L_c} M_{c,l} \right). \tag{1}$$

While MRI provides an overall robustness estimate, it does not capture variability across severities.

*Composite Robustness Index (CRI)* counterbalances this by penalizing instability and normalizing to clean-set performance:

$$\text{CRI} = \frac{1}{C} \sum_{c=1}^{C} \left( \frac{\overline{M}_c}{M_{\text{clean}}} \cdot \frac{1}{\log(1 + \sigma_c) + 1} \right). \tag{2}$$

*Conventions for MRI/CRI.* $C$ is the number of corruption types; $L_c$ is the number of severity levels for corruption $c$. $M_{c,l}$ denotes the task metric evaluated at level $l$ of corruption $c$; $\overline{M}_c$ and $\sigma_c$ are,

Table 1: Comparison of HOI detection methods on HICO-DET and V-COCO under clean and corrupted settings. Results on HICO-DET are reported on Full, Rare and Non-Rare splits; V-COCO results use $\text{AP}^{\#1}_{\text{role}}$ and $\text{AP}^{\#2}_{\text{role}}$. Columns Backb., Det., VLM indicate whether a model uses a pretrained backbone, a detection-model pretraining, or vision–language pretraining. MRI and CRI quantify robustness under corruptions (higher is better).

| Method | Backbone | Pretraining | | | HICO-DET (mAP) | | | | | V-COCO ($\text{AP}_{\text{role}}$) | | | |
| | | Backb. | Det. | VLM | Full | Rare | Non-Rare | MRI↑ | CRI↑ | AP#1 | AP#2 | MRI↑ | CRI↑ |
|---|---|---|---|---|---|---|---|---|---|---|---|---|---|
| **Two-Stage methods** | | | | | | | | | | | | | |
| UPT Zhang et al. (2022a) | R50 | ✓ | ✓ | ✗ | 31.66 | 25.94 | 33.36 | 17.77 | 0.23 | 58.96 | 64.47 | 35.92 | 0.15 |
| GEN-VLKT-S Liao et al. (2022) | R50 | ✓ | ✓ | ✗ | 33.75 | 29.25 | 35.10 | 17.01 | 0.22 | 62.41 | 64.46 | 34.69 | 0.19 |
| UPT Zhang et al. (2022a) | R101 | ✓ | ✓ | ✗ | 32.31 | 28.55 | 33.44 | 17.44 | 0.24 | 60.70 | 66.20 | 33.80 | 0.14 |
| GEN-VLKT-L Liao et al. (2022) | R101 | ✓ | ✓ | ✗ | 34.95 | 31.18 | 36.08 | 18.93 | 0.23 | 63.58 | 65.93 | 37.45 | 0.20 |
| **One-Stage methods** | | | | | | | | | | | | | |
| QPIC Tamura et al. (2021) | R50 | ✓ | ✗ | ✗ | 29.07 | 21.85 | 31.23 | 13.13 | 0.21 | 58.80 | 60.98 | 30.04 | 0.19 |
| CDN-B Zhang et al. (2021a) | R50 | ✓ | ✗ | ✗ | 31.78 | 27.55 | 33.05 | 15.41 | 0.22 | 62.29 | 64.42 | 34.30 | 0.19 |
| DiffHOI Yang et al. (2023) | R50 | ✓ | ✗ | ✗ | 34.41 | 31.07 | 35.40 | 21.09 | 0.23 | – | – | – | – |
| SOV-STG-S Chen et al. (2023) | R50 | ✓ | ✗ | ✗ | 33.80 | 29.28 | 35.15 | 16.94 | 0.22 | 62.88 | 64.47 | 34.40 | 0.20 |
| MUREN Kim et al. (2023) | R50 | ✓ | ✗ | ✗ | 32.88 | 28.70 | 34.13 | 16.32 | 0.23 | 68.10 | 69.88 | 37.10 | 0.18 |
| QPIC Tamura et al. (2021) | R101 | ✓ | ✗ | ✗ | 29.90 | 23.92 | 31.69 | 14.45 | 0.22 | 58.27 | 60.74 | 32.05 | 0.19 |
| CDN-L Zhang et al. (2021a) | R101 | ✓ | ✗ | ✗ | 32.07 | 27.19 | 33.53 | 17.05 | 0.24 | 63.91 | 65.89 | 37.65 | 0.20 |
| FGAHOI Ma et al. (2024a) | Swin-T | ✓ | ✗ | ✗ | 29.81 | 22.17 | 32.09 | 16.05 | 0.22 | – | – | – | – |
| QAHOI Chen & Yanai (2023) | Swin-T | ✓ | ✗ | ✗ | 28.47 | 22.44 | 30.27 | 15.07 | 0.25 | 58.23 | 58.70 | 29.33 | 0.20 |
| RLIPv2 Yuan et al. (2023) | Swin-T | ✓ | ✗ | ✓ | 38.60 | 33.66 | 40.07 | **24.58** | 0.28 | 68.83 | 70.76 | 45.05 | 0.24 |
| SAMPL (ours) | Swin-T | ✓ | ✗ | ✓ | **39.16** | **33.89** | **40.73** | 24.25 | **0.29** | **69.04** | **71.13** | **48.83** | **0.27** |

respectively, the mean and standard deviation of $\{M_{c,l}\}_{l=1}^{L_c}$; $M_{\text{clean}}$ is the clean-set score. Unless otherwise specified, we instantiate $M$ as **mAP (Full)** on HICO-DET and $\text{AP}^{\#2}_{\text{role}}$ on V-COCO.

By combining absolute accuracy with stability across severities, CRI favors models that retain consistent performance under progressively challenging corruptions, complementing the mean-focused view provided by MRI.

## 4 METHOD

In this section, we present our novel Semantic-Aware Masking-based Progressive Learning (SAMPL), aiming to enhance model robustness by progressively guiding models from holistic feature reliance towards robust contextual inference. Unlike standard augmentation methods, SAMPL employs adaptive semantic masking guided by SAM Kirillov et al. (2023), dynamically adjusting masking severity based on validation performance. This structured semantic-level masking strategy explicitly promotes robust contextual reasoning by training models to infer interactions from progressively restricted semantic regions, thus effectively improving general robustness against a variety of realistic disturbances. Moreover, SAMPL's structured progressive curriculum inherently regularizes feature representation, effectively reducing overfitting to dataset biases, thereby further improving generalization under unforeseen corruptions.

**SAM Guided HOI-Aware Semantic Masking (SAMSM).** To fully leverage clean training data, we generate instance-aware masks based on ground-truth bounding boxes and SAM-derived fine-grained masks ($\mathbf{M}_b$). Random edge expansions are applied to prevent strict reliance on instance contours, selectively masking semantic regions while preserving contextual cues. During training, masked pixels are set to zero, compelling the model to infer interactions from partial visual information.

We define various severity levels for the mask, represented by $\Omega = \{\omega_1, \omega_2, \omega_3, \omega_4\}$, corresponding to clean, low, middle, and high levels. Clean level $\omega_1$ employs unmasked data for holistic feature learning, while other levels apply increasing occlusion severity via controlled cover ratios $\left[r_\omega^w, r_\omega^h\right]$. Each semantic mask $\mathbf{M}_\omega^b$ is computed as:

$$\mathbf{M}_\omega^b = \Phi(ConvHull(Dilation(\mathbf{M}_b)), \left[r_\omega^w, r_\omega^h\right]), \tag{3}$$

where dilation and convex hull operations refine the mask shape, and $\Phi$ denotes resizing. This structured masking approach not only simulates realistic semantic occlusions but also inherently promotes context-aware reasoning, significantly enhancing robustness against various semantic and structural corruptions.

---

**Algorithm 1** SAMPL

---

**Input:** Severity levels of semantic aware masking: $\Omega = \{\omega_1, \omega_2, \omega_3, \omega_4\}$, initial threshold: $\tau_{\text{init}}$, maximum epochs: $T$, HOI detection model: $\mathcal{H}_\theta$.
**Initialize:** $N(\Omega) \leftarrow \mathbf{1}^{4 \times 1}$, $\Delta Q_{\text{max}} \leftarrow -\infty$, $p \leftarrow 2$.

1: **for** $t = 1$ to $T$ **do**
2:     Compute $Q_\omega(t)$ for $\omega \in \{\omega_1, \omega_p\}$ using $\mathcal{H}_\theta$.
3:     Get $S_{\omega_p}(t) = \sum_i N(\omega_i) \cdot Q_{\omega_p}(t)$. Get $S_{\omega_1}(t) = N(\omega_1) \cdot Q_{\omega_1}(t)$.
4:     Select severity level $\omega^* = \arg\min_{\omega \in \{\omega_1, \omega_p\}} S_\omega(t)$.
5:     **if** $\omega^* \neq \omega_1$ **then**
6:         Compute $\Delta Q(t) = \frac{Q_{\omega_p}(t) - Q_{\omega_p}(t-1)}{Q_{\omega_p}(t-1)}$.
7:         Update $\Delta Q_{\text{max}} = \max(\Delta Q_{\text{max}}, |\Delta Q(t)|)$.
8:         Compute $\tau(t) = \tau_{\text{init}} \cdot \frac{\Delta Q_{\text{max}}}{|\Delta Q(t)| + \epsilon}$.
9:         **if** $|\Delta Q(t)| < \tau(t)$ and $p < 4$ **then**
10:            $p := p + 1$. Train model on $\omega_p$.
11:            $N(\omega_p) := N(\omega_p) + 1$. $\Delta Q_{\text{max}} \leftarrow -\infty$
12:        **else** Train model on $\omega^*$. $N(\omega^*) := N(\omega^*) + 1$.
13:        **end if**
14:    **else** Train model on $\omega_1$. $N(\omega_1) := N(\omega_1) + 1$.
15:    **end if**
16: **end for**

---

**Score Guided Progressive Feature Learning.** To effectively utilize SAM-guided masks, we design a novel score-guided progressive learning framework that incorporates a frequency memory bank across severity levels.

Initially, training begins with the lowest severity level, while a memory bank is designed to track the utilization frequency of different severity levels. The perturbation level is then progressively increased based on the relative change in the evaluation metric between consecutive epochs. Notably, the severity level is only upgraded when the selected level differs from $\omega_1$. Let $\Delta Q(t)$ represent the relative change in the evaluation score on the validation set at epoch $t$:

$$\Delta Q(t) = \frac{Q_\omega(t) - Q_\omega(t-1)}{Q_\omega(t-1)}, \tag{4}$$

where $Q_\omega(t-1)$ is the evaluation metric for the current masking severity level in the previous epoch. To ensure stability, the perturbation level is upgraded when the magnitude of $\Delta Q(t)$ falls below a dynamically adjusted threshold $\tau(t)$:

$$\tau(t) = \tau_{\text{init}} \cdot \frac{\Delta Q_{\text{max}}}{|\Delta Q(t)| + \epsilon}, \tag{5}$$

where $\tau_{\text{init}}$ is the initial threshold, $\Delta Q_{\text{max}} = \max_{k \leq t} |\Delta Q(k)|$ is the maximum observed relative change in $Q$ up to epoch $t$, and $\epsilon$ is a small constant to avoid division by zero.

The severity level $l$ is updated as:

$$p := \begin{cases} p + 1 & \text{if } |\Delta Q(t)| < \tau(t) \text{ and } p < 4, \\ p & \text{otherwise.} \end{cases} \tag{6}$$

At epoch $t$, let $N(\omega)$ represent the number of times one severity level $\omega$ has been selected up to epoch $t$ to serve as the aforementioned frequency memory bank. Let $Q_\omega(t)$ denote the validation performance for severity level $\omega$ at epoch $t$. The severity level score $S_\omega(t)$ is thereby defined as:

$$S_{\omega_p}(t) = \sum_i N(\omega_i) \cdot Q_{\omega_p}(t), \omega_i \in \Omega/\{\omega_1\}. \tag{7}$$

Then we obtain $S_{\omega_1} = N(\omega_1) \cdot Q_{\omega_1}(t)$. At each epoch, the next training severity level is chosen between the original data $\omega_1$ (*i.e.*, lowest severity level to achieve holistic feature learning) and $\omega_p$. The selected severity level $\omega^*$ is determined as:

$$\omega^* = \arg\min_{\omega \in \{\omega_1, \omega_p\}} S_\omega(t). \tag{8}$$

The whole procedure is summarized in Alg. 1.

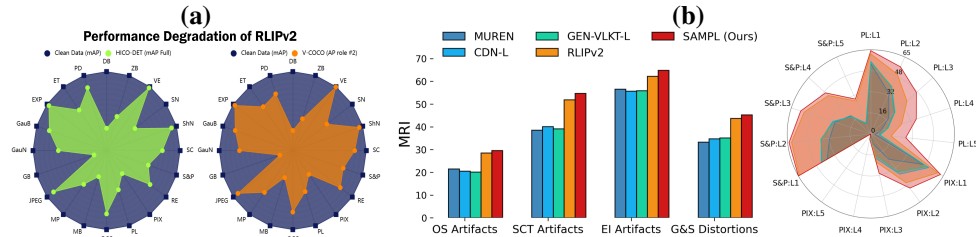

Figure 2: (a) Performance degradation of RLIPv2 Yuan et al. (2023) on HICO-DET and V-COCO under RoHOI. The left radar chart shows the relative mAP (Full) drop on HICO-DET, while the right shows the decline in $AP^{\#2}_{role}$ on V-COCO. Clean dataset performance serves as a reference, with shaded areas indicating corruption impact. (b) Comparison of SAMPL with strong two-stage and one-stage baselines trained on V-COCO under RoHOI benchmarks. The left bar chart groups performance by corruption categories, while the right radar chart compares models across five severity levels for three corruptions. Corruptions are denoted as Abbreviation, *i.e.*, $L\gamma$, where $\gamma$ represents the severity level. Details are shown in Sec. 3 and Appendix.

## 5 EXPERIMENTAL RESULTS

**Implementation Details.** Our SAMPL method is built upon RLIPv2 Yuan et al. (2023) with a Swin-Tiny Liu et al. (2021) backbone and has $214M$ trainable parameters. The backbone is pretrained on Visual Genome Krishna et al. (2017), MS-COCO Lin et al. (2014) and Object365 Shao et al. (2019), and fine-tuned on the HICO-DET or V-COCO datasets. We use the AdamW optimizer Loshchilov & Hutter (2019) with an initial learning rate of $1e$-4 (backbone and text encoder at $1e$-5), linear warmup followed by cosine decay, weight decay of $1e$-4, and batch size 2 per GPU with gradient accumulation over 2 steps. Models are trained for 80 epochs using cross-entropy loss for object classification, focal loss for verb classification, and L1 + GIoU loss for bounding, with KL divergence for latent space regularization. The encoder and decoder consist of 6 and 3 layers, respectively. For the dynamic severity adjustment mechanism, we set the initial threshold as $\tau_{init} = 0.15$ and $\epsilon = 10^{-6}$.

**Analysis of the Benchmark.** Tab. 1 presents the performance comparison between SAMPL and existing one-stage and two-stage HOI detection methods on HICO-DET Wang et al. (2020) and V-COCO Gupta & Malik (2015) datasets. SAMPL surpasses RLIPv2 Yuan et al. (2023), improving MRI by 3.78 and CRI by 12.5% on V-COCO, demonstrating superior robustness against visual degradations. On HICO-DET, SAMPL achieves a 3.6% higher CRI while maintaining comparable MRI, ensuring strong generalization without compromising clean-set accuracy. These results are further supported by Fig. 2 (b), where SAMPL consistently outperforms two-stage and one-stage baselines across all 4 corruption categories in the RoHOI benchmark, achieving the highest MRI in each group and demonstrating improvements over the current SOTA model at every severity level (detailed in the Appendix). While extreme noise levels still pose challenges, SAMPL significantly improves robustness compared to prior methods by progressively training the model with semantic-aware occlusions, enabling it to learn context-driven features that are resilient to diverse and severe corruptions. These results highlight SAMPL's effectiveness in enhancing HOI detection under corrupted conditions through our proposed semantic-aware masking and progressive learning. Through a comprehensive analysis of representative HOI models, we derive the following key findings.

*Observation 1: Vision–language relational pretraining is associated with higher corruption-robustness.* Within each dataset (Tab. 1), HOI detectors trained with relational vision–language objectives occupy the top tier of MRI/CRI, while transformer-based counterparts without such cross-modal relational supervision lag behind. A plausible mechanism is that relational fusion during pretraining aligns verb–object evidence and regularizes interaction features against distribution shift; We interpret this pattern as associative rather than causal; backbone capacity remains a potential confounder.

*Observation 2: Backbone scaling yields architecture-dependent effects; association/decoding design mediates robustness.* Table 1 shows a divergence: enlarging the backbone reduces MRI/CRI for a query-based two-stage pipeline (UPT, R50→R101), whereas scaling a guided-association two-

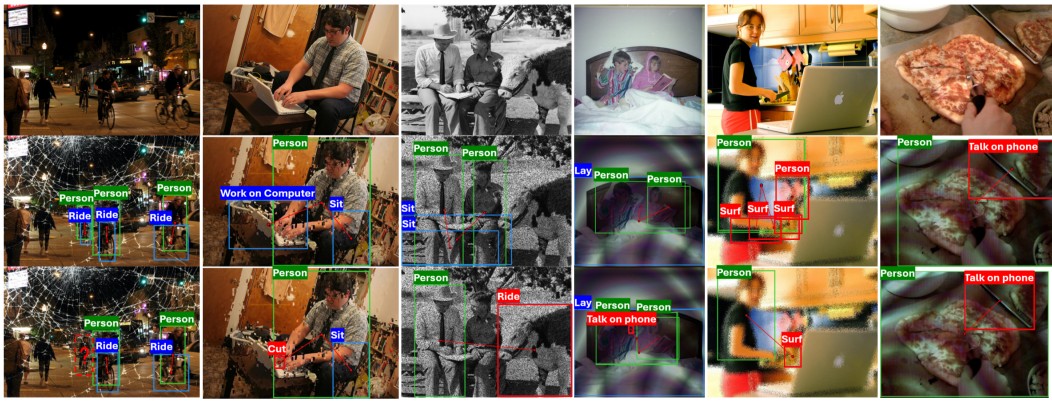

Figure 3: Qualitative comparison of HOI detection under corruption between SAMPL (ours) and RLIPv2 Yuan et al. (2023). The first row shows uncorrupted images, while the second and third rows present SAMPL and RLIPv2 results on corrupted images. Missed predictions are in dashed red, false interactions in red, and correct ones in blue.

Table 2: Ablation studies on V-COCO. (a) Ablation on various perturbation strategies during training: SAM-guided HOI-aware Semantic Masking (SAMSM), Adaptive Gaussian Noise Perturbation (AGNP), and Randomized Pixelation Perturbation (RPP). (b) Ablation on training with different severity level combinations for SAMSM: each instance is masked based on 40%–80% of its bounding box height and width, using three candidate levels during training.

**(a) Training perturbation strategies**

| Perturbation Type | V-COCO | | MRI⇑ | CRI⇑ |
|---|---|---|---|---|
| | AP#1_role | AP#2_role | | |
| RLIPv2+SAMSM | **68.27** | 70.24 | **48.20** | **0.27** |
| RLIPv2+AGNP | 68.33 | **70.51** | 44.74 | 0.24 |
| RLIPv2+RPP | 67.69 | 69.64 | 47.59 | 0.25 |

**(b) SAMSM cover ratio combinations**

| Cover Ratios | V-COCO | | MRI⇑ | CRI⇑ |
|---|---|---|---|---|
| | AP#1_role | AP#2_role | | |
| 40%/50%/60% | **69.04** | **71.13** | **48.83** | 0.27 |
| 40%/60%/70% | 68.51 | 70.53 | 48.44 | **0.27** |
| 50%/60%/80% | 68.21 | 70.29 | 48.26 | **0.27** |
| 50%/70%/80% | 68.52 | 70.45 | 45.72 | 0.22 |

stage framework (GEN-VLKT, S→L) improves both metrics. Robustness under corruptions therefore depends more on how human–object evidence is formed and decoded. Representative mechanisms include guided embeddings and decoupled instance/interaction decoding, which outweigh the effect of model size alone.

*Observation 3: Clean-set accuracy and robustness are partially decoupled; stability is a distinct axis captured by CRI.* Multiple entries in Tab. 1 attain strong clean $AP_{role}$ yet only mid-range CRI, whereas others with comparable clean scores exhibit higher CRI. This mirrors corruption-benchmarking results emphasizing that mean accuracy and stability under corruptions probe different properties of the representation; CRI operationalizes the latter by rewarding low inter-severity variance and complements MRI's mean view.

*Observation 4: Progressive semantic masking primarily improves stability (CRI) and smooths severity–performance curves.* Relative to a strong vision–language pretrained baseline in Tab. 1, SAMPL consistently increases CRI within each dataset, while MRI gains are moderate or dataset-dependent. This pattern aligns with evidence that self-attentive encoders are sensitive to patch-level corruptions: structure-aware, progressive masking encourages attention to retain relational evidence under partial visibility, yielding smoother severity–performance profiles without sacrificing clean accuracy.

**Analysis of the Ablation Study.** To investigate the impact of perturbation-based training on model robustness, we conduct a series of ablation experiments on the V-COCO dataset. The results, presented in Tab. 2, analyze the effects of different perturbation types and perturbation level settings on model performance.

*Ablation of the Semantic-Aware Masking:* Tab. 2 (a) evaluates three perturbation strategies applied independently during training, excluding our progressive learning method to isolate their direct impact. *SAM guided HOI-aware Semantic Masking (SAMSM)* is applied in 50% of training epochs, occluding instance's regions while preserving contextual cues. *Adaptive Gaussian Noise Perturbation (AGNP)* introduces feature-level noise with a 50% probability, while *Randomized Pixelation*

*Perturbation (RPP)* pixelates input images with the same probability as data augmentation. *SAMSM* achieves the highest MRI and CRI scores, suggesting that structured occlusions enhance robustness by enforcing relational reasoning under missing object regions. Unlike *AGNP*, which perturbs latent features without spatial structure, *SAMSM* provides localized challenges that align with real-world occlusions. Despite expectations that *RPP* would improve robustness—given that *pixelation* causes the most significant performance drop in baseline corruption tests (as shown in Fig. 2 (a))—its randomized nature indiscriminately distorts both essential and redundant information, making adaptation less effective. These results suggest that robustness in HOI detection benefits more from structured perturbations that challenge object-awareness rather than indiscriminate degradations that introduce confounding artifacts.

*Effect of Perturbation Levels:* Tab. 2 (b) analyzes the impact of masking severity on robustness. *SAMSM* occludes $40\%$–$80\%$ of each instance's bounding box, with the $40\%/50\%/60\%$ setting achieving the best trade-off, maximizing MRI and CRI while maintaining competitive AP scores. Moderate occlusions enhance feature learning via contextual cues, while excessive masking ($> 70\%$) disrupts spatial structures, reducing discriminative capacity. These findings emphasize the need for controlled occlusions that challenge models without excessive information loss.

Overall, perturbation-based training enhances HOI detection robustness, with SAMSM proving most effective by balancing global context preservation and local feature resilience. Moderate perturbation achieves optimal robustness without sacrificing clean-data accuracy.

**Analysis of the Qualitative Results.** Fig. 3 compares SAMPL with RLIPv2 under various corruptions, demonstrating SAMPL's superior robustness in the first four cases while highlighting shared failure patterns in the last two. Under *Screen Crack* corruption (Col. 1), SAMPL accurately detects occluded background persons and their "ride" interactions, whereas RLIPv2 misses smaller subjects, indicating SAMPL's stronger resilience to fragmented textures, as SAMPL uses semantic-aware masking that selectively occludes HOI-relevant regions, training the model to recognize interactions from partial and degraded visual cues. In *Pixelation* noise (Col. 2), our method correctly recognizes "work on computer", while RLIPv2 misclassifies it as "cut", suggesting greater robustness against structured perturbations. Under *Salt-and-Pepper Noise* (Col. 3), RLIPv2 confuses "sit" with "ride", likely due to the presence of an adjacent animal, whereas SAMPL maintains stable interaction recognition, reducing reliance on object-centric biases. In *Moire Pattern* (Col. 4) corruption, RLIPv2 incorrectly detects "talk on phone", possibly due to moire-induced distortions interfering with high-frequency texture recognition. However, under high-severity *Glass Blur* (Col. 5) and *Moire Pattern* corruption (Col. 6), both models misclassify actions as "surf" or "talk on phone", potentially because V-COCO's limited action categories encourage overfitting to common interactions under extreme degradation rather than a traditional long-tail distribution issue. These results highlight SAMPL's advantages in structured occlusions and texture-based corruptions. However, both models remain vulnerable under severe degradations, where dataset priors dominate predictions.

## 6 CONCLUSION

In this work, we introduced RoHOI, the first benchmark dedicated to evaluating the robustness of Human-Object Interaction (HOI) detection models under 20 diverse real-world corruptions. Our analysis revealed that even state-of-the-art approaches degrade substantially under perturbations, underscoring the need for robustness-focused evaluation. To address this challenge, we proposed SAMPL (Semantic-Aware Masking-based Progressive Learning), which leverages SAM-generated region-aware masks and a dynamic progressive masking strategy to strengthen resilience. Extensive experiments on HICO-DET and V-COCO showed that SAMPL consistently outperforms strong baselines under corruption while preserving state-of-the-art performance on clean benchmarks. Beyond performance gains, our results highlight that clean accuracy and robustness are only partially coupled, that structured perturbations provide more effective robustness cues than random noise, and that architecture design plays a crucial role in stability. While RoHOI captures a broad range of corruptions, it still focuses on synthetic perturbations; future work should explore robustness under real-world domain shifts, extreme degradations, and fairness-aware evaluation. We envision RoHOI as a catalyst for advancing robustness research in HOI detection, and SAMPL as an initial step toward building reliable and deployable models for safety-critical applications.

ETHICS STATEMENT

This work introduces both a methodological contribution and a new benchmark for robust HOI. All experiments are performed exclusively on publicly available datasets from the HOI research community, whose ethical standards have been validated through publication in leading venues. The research poses no concerns related to privacy, security, fairness, bias, or harmful applications, and fully adheres to established principles of research integrity and ethical practice.

REPRODUCIBILITY STATEMENT

The source code of our proposed approach is available at https://anonymous.4open.science/r/RoHOI-365E/ to ensure the reproducibility.

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

# A  APPENDIX

## A.1  SOCIAL IMPACT AND LIMITATIONS

**Social Impact:** We introduce RoHOI, the first benchmark for assessing HOI detection robustness under real-world corruptions, incorporating 20 degradation types to bridge the gap between theoretical advancements and deployment. Robust HOI detection is crucial for safety-critical applications, yet existing models struggle with occlusions and noise. To address this, we propose SAMPL, a semantic-aware masking and progressive learning approach that enhances resilience to distortions, outperforming SOTA models in both clean and corrupted settings. However, challenges remain, including biased predictions and disparities across interaction categories, underscoring the need for further research in bias mitigation, domain adaptation, and fairness-aware robustness.

**Limitations:** RoHOI focuses on synthetic corruptions but may not fully capture real-world domain shifts like scene-specific occlusions and adversarial attacks, highlighting the need for real-world corrupted data. While SAMPL improves robustness, it still struggles with severe distortions (*e.g.*, moire patterns, glass blur), indicating the need for further architectural advancements.

## A.2  STANDARD HOI EVALUATION METRICS

This section provides a detailed explanation of the evaluation metrics used for HICO-DET Xu et al. (2019) and V-COCO Gupta & Malik (2015) in our main paper. These include the standard mAP metrics for HICO-DET and $AP_{role}$ variations for V-COCO.

### A.2.1  **mAP**: FULL, RARE, NON–RARE

We evaluate models on HICO-DET using the standard mAP under an IoU threshold of 0.5. The reported metrics include mAP (Full), which evaluates all 600 HOI categories; mAP (Rare), assessing performance on low-frequency categories with fewer than 10 instances; and mAP (Non-Rare), focusing on categories with at least 10 instances.

### A.2.2  $\mathbf{AP}^{\#1}_{\mathbf{ROLE}}$ AND $\mathbf{AP}^{\#2}_{\mathbf{ROLE}}$

We evaluate V-COCO using $AP_{\mathbf{role}}$, which measures interaction detection accuracy. $AP^{\#1}_{\mathbf{role}}$ assesses action recognition without strict role object localization. $AP^{\#2}_{\mathbf{role}}$ requires both action and object localization, with an IoU threshold of 0.5.

## A.3  CORRUPTION TYPES IN THE DATASET

Building upon the general introduction in the main text, this section provides a more detailed breakdown of the 20 corruption types used in our RoHOI benchmark.

### A.3.1  OPTICAL SYSTEM-INDUCED (OS) ARTIFACTS

**Motion Blur (MB):** Blur caused by the relative motion between the camera and the scene during exposure. This artifact is included as it commonly occurs in dynamic scenes or when handheld cameras are used. Severity levels are generated by increasing the kernel size of a linear motion blur filter. (Details shown in Figure 4, row 1.)

**Defocus Blur (DB):** Blur resulting from the lens being improperly focused on the subject. This noise reflects the limitations of camera focus systems, particularly in low-light or high-speed scenarios. Severity levels are controlled by varying the radius of a circular point spread function (PSF). (Details shown in Figure 4, row 2.)

**Gaussian Blur (GauB):** Blur due to scattering or inherent optical limitations of the lens. This artifact simulates the diffusion of light caused by imperfect optical surfaces or aperture settings. Severity levels are adjusted by increasing the standard deviation of the Gaussian kernel. (Details shown in Figure 4, row 3.)

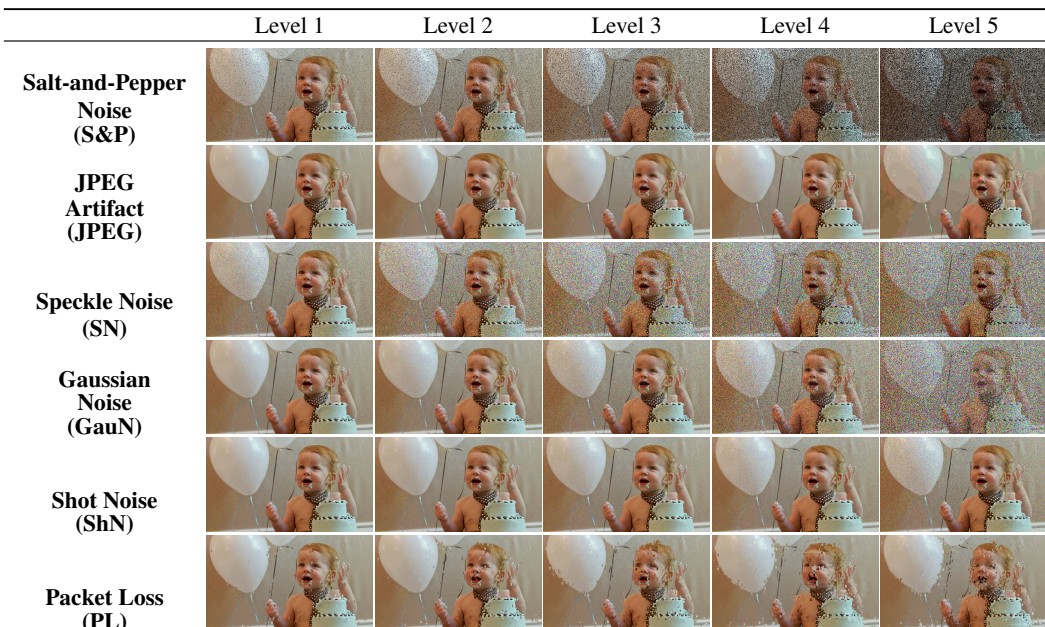

Figure 4: Visual examples of Optical System (OS)-Induced artifacts across severity levels.

Figure 5: Visual examples of Sensor, Compression and Transmission (SCT) artifacts across severity levels.

**Glass Blur (GB):** Blur introduced when imaging through imperfect or contaminated transparent materials (*e.g.*, glass or plastic). This type is included to represent common real-world challenges such as photographing through windows or protective covers. Severity levels are generated by applying iterative small displacements and local blurring. (Details shown in Figure 4, row 4.)

### A.3.2 SENSOR, COMPRESSION AND TRANSMISSION (SCT) ARTIFACTS

**Salt-and-Pepper Noise (S&P):** Discrete black-and-white pixel noise caused by sensor malfunction or transmission errors. This represents rare but impactful noise patterns in hardware-limited imaging systems. Severity levels are generated by increasing the probability of random pixel corruption. (Details shown in Figure 5, row 1.)

**JPEG Artifacts (JPEG):** Blocky or blurry artifacts resulting from lossy image compression, which are used to evaluate model robustness to low-quality image formats frequently encountered in data storage and transmission. Severity levels are controlled by progressively reducing the JPEG quality factor. (Details shown in Figure 5, row 2.)

|  | Level 1 | Level 2 | Level 3 | Level 4 | Level 5 |
|---|---|---|---|---|---|

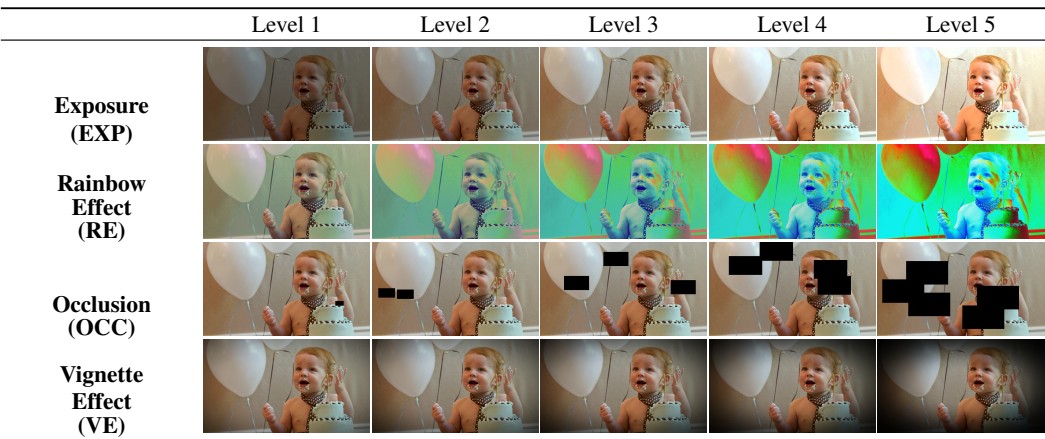

**Exposure (EXP)**

**Rainbow Effect (RE)**

**Occlusion (OCC)**

**Vignette Effect (VE)**

Figure 6: Visual examples of Environmentally Induced (EI) artifacts across severity levels.

**Speckle Noise (SN):** Speckle noise resembles the grainy texture seen on old printed photos or low-quality digital images taken in dim lighting. At lower severity levels, it appears as faint specks of dust on a camera lens, barely noticeable unless viewed closely. As severity increases, the noise becomes more pronounced, resembling the rough texture of sandpaper or the graininess of a poorly printed newspaper. At the highest severity levels, the effect is akin to heavy rain blurring a car windshield, significantly degrading image clarity. (Details shown in Figure 5, row 3.)

**Gaussian Noise (GauN):** Additive noise caused by sensor imperfections such as thermal noise or quantization errors. This is a ubiquitous noise model, reflecting baseline sensor limitations in most imaging devices. Severity levels are determined by progressively increasing the noise standard deviation. (Details shown in Figure 5, row 4.)

**Shot Noise (ShN):** Random variations in pixel intensity caused by photon-counting statistics during exposure. This noise type represents the fundamental physical limits of image sensors in low-light conditions. Severity levels are controlled by adjusting the scale of the Poisson-distributed noise. (Details shown in Figure 5, row 5.)

**Packet Loss (PL):** Missing or distorted image regions caused by data corruption during transmission. This noise simulates real-world challenges in wireless or network-based imaging systems. Severity levels are generated by increasing the number and size of corrupted or duplicated regions. (Details shown in Figure 5, row 6.)

### A.3.3 Environmentally Induced (EI) Artifacts

**Exposure Artifacts (EXP):** Loss of image details caused by underexposure or overexposure due to extreme lighting conditions. Higher severity levels increase the intensity of overexposure (bright regions saturate) or underexposure (dark regions lose details), leading to stronger visual degradation. This type is included to represent real-world challenges in scenes with poor or uncontrolled lighting. (Details shown in Figure 6, row 1.)

**Rainbow Effect (RE):** Typically caused by the interaction between the projection system's optical components (*e.g.*, color wheels in DLP projectors) and the camera sensor. As severity increases, the color distortion becomes more pronounced, with broader and more intense rainbow bands appearing across the image, significantly altering local color distribution. This noise enables the evaluation of model robustness against chromatic distortions, particularly in scenes involving projected displays and reflective surfaces. (Details shown in Figure 6, row 2.)

**Occlusion (OCC):** Partial obstruction of the field of view by external objects. Increasing severity levels introduce randomly placed black rectangular occlusions of varying sizes, progressively removing more contextual information from the image. This is a critical inclusion as occlusions are prevalent in crowded or dynamic environments. (Details shown in Figure 6, row 3.)

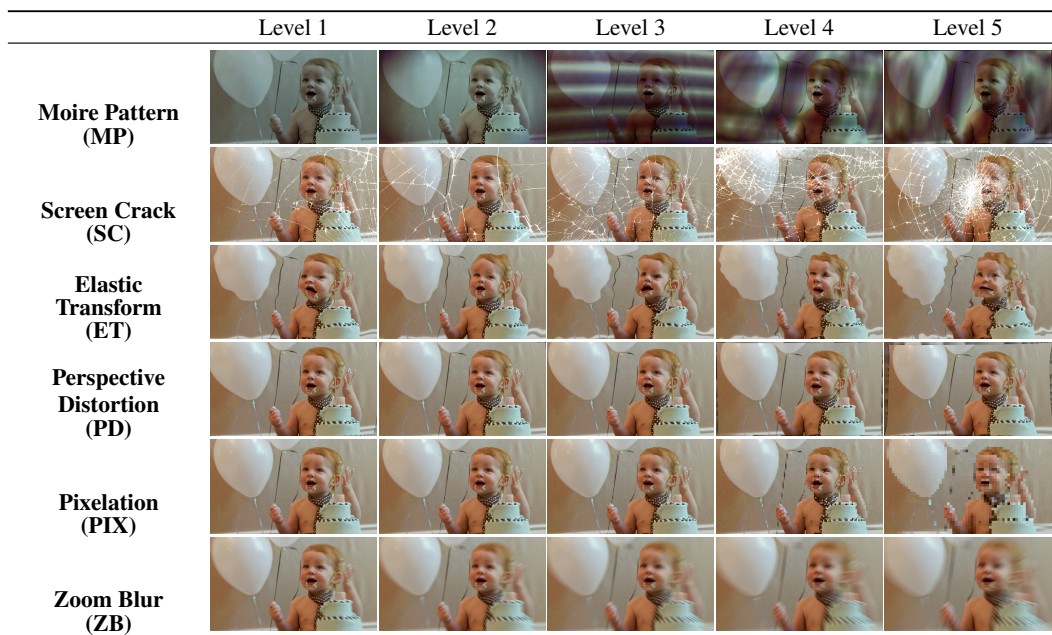

|  | Level 1 | Level 2 | Level 3 | Level 4 | Level 5 |

Figure 7: Visual examples of Geometric and Scene (G&S) Distortions across severity levels.

**Vignette Effect (VE):** Gradual darkening at the edges of an image caused by lens design or aperture settings. With higher severity, the vignette effect has darker and wider peripheral regions, increasingly suppressing details in the outer parts of the image. This type represents a common optical limitation that affects peripheral details in wide-field imaging. (Details shown in Figure 6, row 4.)

### A.3.4   GEOMETRIC AND SCENE (G&S) DISTORTIONS

**Moire Pattern (MP):** Interference patterns resulting from fine texture overlap with the camera's pixel grid (e.g., fabric or screens). This artifact captures the challenges posed by imaging repetitive textures or high-frequency patterns. Severity levels adjust the density and contrast of the interference patterns. (Details shown in Figure 7, row 1.)

**Screen Crack Artifacts (SC):** Visual obstructions or patterns caused by photographing through cracked or damaged surfaces. This type is included to reflect realistic scenarios where images are captured through broken screens or fractured glass. Higher severity levels introduce progressively denser and larger crack patterns, further obscuring the image. (Details shown in Figure 7, row 2.)

**Elastic Transform (ET):** Non-linear distortions caused by perspective changes, material deformations, or uneven air density. This artifact represents challenges encountered in scenes with flexible or moving objects. Higher severity levels correspond to greater displacement magnitudes, leading to noticeable warping. (Details shown in Figure 7, row 3.)

**Perspective Distortion (PD):** This artifact simulates geometric distortions from lens imperfections, such as barrel and pincushion distortion. Severity is increased by applying progressively stronger transformation matrices, exaggerating the curvature and altering the perceived structure of objects. (Details shown in Figure 7, row 4.)

**Pixelation (PIX):** Blocky appearance due to low resolution or excessive digital zoom. This represents a key limitation in devices with constrained sensor capabilities or in images subjected to aggressive downscaling. Higher severity levels result in coarser pixel blocks, significantly reducing image detail. (Details shown in Figure 7, row 5.)

**Zoom Blur (ZB):** Radial blur caused by rapid changes in focal length during image capture. Zoom blur is simulated by iteratively applying slight scaling transformations and averaging the results to create a progressive radial blur effect. Higher severity levels increase both the scaling intensity and the number of blended layers, leading to a stronger blur radiating outward from the center. This

Table 3: Comparison of MRI scores between our SAMPL method and the best-performing one-stage and two-stage HOI detection models on the RoHOI benchmark. The table presents results across 20 corruption types, evaluating model robustness under diverse corruptions. Corruption abbreviations follow the definitions provided in the main paper.

| Model | MB | DB | GauB | GB | GauN | ShN | S&P | JPEG | SN | PL | EXP | RE | OCC | VE | MP | SC | ET | PD | PIX | ZB |
|---|---|---|---|---|---|---|---|---|---|---|---|---|---|---|---|---|---|---|---|---|
| CDN-L Zhang et al. (2021a) | 15.38 | 10.12 | 48.73 | 7.56 | 41.67 | 60.92 | 28.78 | 56.89 | 24.24 | 28.18 | 64.67 | 41.06 | 49.75 | 67.07 | 30.67 | 44.30 | 37.33 | 50.79 | 22.39 | 22.45 |
| GEN-VLKT-L Liao et al. (2022) | 14.64 | 9.90 | 48.27 | 7.74 | 40.50 | 60.63 | 28.37 | 56.31 | 20.40 | 28.52 | 64.94 | 40.48 | 50.48 | 67.48 | 32.42 | 44.21 | 38.27 | 51.15 | 21.11 | 23.24 |
| MUREN Kim et al. (2023) | 15.71 | 12.10 | 51.08 | 6.77 | 41.65 | 62.59 | 26.08 | 56.32 | 20.46 | 23.89 | 66.65 | 38.55 | 51.78 | 69.12 | 25.63 | 44.36 | 38.79 | 52.34 | 15.97 | 22.22 |
| RLIPv2-ParSeDA Yuan et al. (2023) | 25.31 | 17.84 | 57.82 | 12.99 | 50.91 | 66.72 | 50.58 | 64.68 | 40.11 | 38.34 | 68.39 | 55.99 | 54.61 | 69.82 | 45.93 | 59.14 | 48.14 | 53.01 | 26.12 | 29.73 |
| SAMPL (Ours) | **26.63** | **19.03** | **58.56** | **13.87** | **52.46** | **68.06** | **52.66** | **65.81** | **42.24** | **46.69** | **69.22** | **57.31** | **62.04** | **70.77** | **47.89** | **60.72** | **48.99** | **53.21** | **29.90** | **30.49** |

method ensures a smooth and natural-looking zoom effect without abrupt artifacts. (Details shown in Figure 7, row 6.)

## A.4 PURPOSE OF INCLUSION

These noise types are selected to comprehensively represent real-world scenarios that affect image quality, ranging from optical and sensor limitations to environmental and dynamic scene factors. Their inclusion ensures the dataset's relevance for evaluating model robustness in diverse and realistic conditions.

## A.5 BENCHMARK RESULTS ON CORRUPTION TYPES

Table 3 presents the MRI scores of our SAMPL method compared to the strongest one-stage and two-stage HOI detection baselines on the RoHOI benchmark. SAMPL consistently outperforms prior methods across all 20 corruption types, achieving the highest scores in every category. Notably, our approach shows significant improvements in handling severe degradations such as pixelation and packet loss, demonstrating enhanced robustness. These results validate the effectiveness of our SAMPL strategy in improving model resilience against real-world corruptions.

## A.6 LLM USAGE STATEMENT

In this work, large language models (LLMs) were used as general-purpose assistant tools to improve writing clarity and grammar. Specifically, LLMs helped draft some introductory text and proofread the paper to enhance readability. All scientific content and research ideas were developed by the authors without direct generation or invention by LLMs. The authors take full responsibility for the integrity and originality of the work.

