# OpenReview forum: "RoHOI: Robustness Benchmark for Human-Object Interaction Detection"
_ICLR.cc/2026/Conference — ICLR 2026 Conference Withdrawn Submission_

### Official Review · Reviewer_G7qP · 2025-10-27

**Soundness:** 2
**Presentation:** 2
**Contribution:** 1
**Rating:** 2
**Confidence:** 5

**Summary:**

This paper addresses a critical and underexplored problem in the field of Human-Object Interaction (HOI) detection: the robustness of existing models against real-world data corruptions. The authors make three core contributions:

1.  **RoHOI Benchmark**: They introduce the first robustness benchmark specifically designed for HOI detection. This benchmark incorporates 20 types of synthetic corruptions at 5 severity levels on two standard datasets (HICO-DET and V-COCO) to systematically evaluate the performance of existing HOI models beyond ideal conditions.
2.  **New Evaluation Metric**: In addition to the standard Mean Robustness Index (MRI), they propose a novel Composite Robustness Index (CRI). The CRI not only measures average performance but also rewards stability by penalizing performance variance across different corruption severities, thus providing a more comprehensive assessment of robustness.
3.  **SAMPL Training Strategy**: To enhance model robustness, this paper proposes a novel method called Semantic-Aware Masking-based Progressive Learning (SAMPL). The method utilizes the Segment Anything Model (SAM) to generate HOI-relevant semantic masks and employs a curriculum learning strategy to dynamically and progressively increase the masking difficulty, compelling the model to learn contextual reasoning from incomplete, partial information.

Experimental results show that existing SOTA models suffer a significant performance drop on the RoHOI benchmark, while the proposed SAMPL method substantially improves model robustness on corrupted data while maintaining high performance on clean datasets.

**Strengths:**

1.  **Pioneering and Important**: The most prominent strength of this paper is the introduction of the **first** comprehensive benchmark, RoHOI, specifically for evaluating the robustness of HOI detection. This is a vital contribution that will benefit the entire research community by enabling standardized assessment and driving progress in model reliability.
2.  **Depth and Breadth of Evaluation**: The newly proposed CRI metric is excellent. It goes beyond merely measuring the average performance drop by incorporating "stability" as a key consideration, making the evaluation of robustness more comprehensive and fair.
3.  **Effectiveness of the Method**: The proposed SAMPL method is not only theoretically sound but also empirically proven to be effective. It consistently outperforms existing SOTA models on multiple benchmarks and metrics, demonstrating its strong performance.
4.  **Insightful Analysis**: The experimental section does more than just list numbers. The authors provide an in-depth analysis of the results and distill several key observations, such as the correlation between VLM pre-training and robustness, and the non-linear relationship between model size and robustness. These findings offer valuable insights for future research.

**Weaknesses:**

1.  **Dependence on Synthetic Corruptions**: The main limitation of this work is its reliance on **algorithmically generated synthetic corruptions**. While this is a common practice in robustness research, synthetic corruptions may not fully capture the complexity and diversity of real-world degradations. This is mentioned in the appendix, but it deserves more prominence in the main text.
2.  **Insufficient Discussion on SAMPL's Generalization Mechanism**: The core mechanism of SAMPL is to simulate "semantic occlusion." The experiments impressively show its effectiveness across all 20 corruption types. However, the paper could delve deeper into the intrinsic mechanism of its generalization: why does training on a task that primarily simulates "loss of spatial information" (occlusion) effectively improve resistance against fundamentally different types of corruption like "color distortion" (Rainbow Effect) or "compression artifacts" (JPEG Artifacts)? Further exploration of the causal link between "semantic occlusion training" and "cross-modal robustness improvement," either theoretically or through visualization, would significantly enhance SAMPL's persuasiveness.
3.  **Hyperparameter Sensitivity of Progressive Learning**: Algorithm 1 introduces several hyperparameters, such as the initial threshold `τ_init`. While the method is effective, readers may be curious about the framework's sensitivity to these hyperparameters. Since the core mechanism relies on a dynamic threshold to decide when to "increase" the occlusion severity, an improper choice of `τ_init` or the update strategy parameters could affect the training pace or even lead to unstable convergence. Supplementing with an analysis of the impact of key hyperparameters (e.g., the trend of MRI/CRI changes within a reasonable range) would help assess SAMPL's robustness and portability and better guide future research in reproducing and improving the method.
4.  **The model's "Semantic Awareness" and "Contextual Reasoning" Capabilities**: This ability should not be limited to resisting pixel-level damage on "seen" categories; it should also be transferable to new scenarios the model has "not seen." The Zero-Shot scenario is the gold standard for testing this kind of capability transfer and generalization. Currently, the paper demonstrates SAMPL's robustness on known categories. However, it is unclear how much of the performance improvement comes from adapting to the data distribution itself (i.e., becoming more adapted to the corrupted dataset) versus learning a generalizable reasoning ability.

---

**Questions:**

1.  Regarding cross-corruption generalization: If the model were trained with SAMPL using only a subset of corruption types (e.g., only blurs and occlusions), could it generalize its robustness to unseen corruption types (e.g., noise or pixelation)? This could further clarify whether the learned robustness is universal or type-specific.
2.  Compared to the proposed dynamic progressive learning framework, how does a simpler strategy perform, such as mixing all severity levels of masked data with fixed probabilities throughout the training? Is the performance gain from the complex scheduling significant enough to justify its design?
3.  The conclusion mentions that future work should explore real-world domain shifts and extreme degradations. Beyond establishing a test set with real-world corrupted data, how do the authors believe the SAMPL framework could be extended at the methodological level to better address these real-world challenges?

---

### Official Review · Reviewer_TS55 · 2025-10-29

**Soundness:** 2
**Presentation:** 3
**Contribution:** 2
**Rating:** 2
**Confidence:** 5

**Summary:**

This paper introduces RoHOI, a new benchmark for evaluating the robustness of Human-Object Interaction detection models.

 The benchmark is constructed by applying a set of 20 algorithmically-generated corruptions to 2 standard HOI datasets. The authors show that some existing HOI models suffer significant performance degradation on this benchmark.

To measure this, they propose a metric, the Composite Robustness Index (CRI).

Finally, to address the performance drop, they propose a training strategy called Semantic-Aware Masking-based Progressive Learning, which uses semantic masks from SAM in a progressive curriculum to improve model resilience.

Experiments show that SAMPL improves robustness scores on RoHOI compared to baseline models.

**Strengths:**

1.  Efforts in evaluating different HOI baselines. The paper evaluates a representative range of existing HOI models, including one-stage, two-stage, and vision-language-based approaches. This provides a valuable snapshot of the current field's weaknesses in the face of corruptions.
2. Writing and clarity. The paper is generally well-written, the language usage is good, and the core ideas are easy to follow.
4. The presentation is good. Figures 1, 3, and the appendix figures clearly illustrate the corruption types and the method's qualitative results.
5. Reproducibility of the proposed method. The authors release the code of the proposed SAMPL method for review.

**Weaknesses:**

1. Benchmark construction seems far from its claimed motivation and contribution. The paper repeatedly claims to evaluate models under "real-world conditions". However, the benchmark is constructed by applying synthetic, algorithmic corruptions to clean, in-distribution datasets. This is a common methodology, but it may not capture the complexity of true real-world degradation, which often involves combinations of corruptions, and most importantly, out-of-distribution and domain-shift challenges (e.g., novel scenes, weather) that are completely ignored here.
2. Weak motivation and justification of the setting. The paper's core motivation seems not so sufficient. While it lists safety-critical applications like autonomous driving and robotics, it fails to provide a concrete, compelling scenario where robust HOI detection (a complex, fine-grained task) is the critical bottleneck on a severely corrupted image. In many such scenarios, a system would likely rely on simpler, more robust sensors (e.g., LiDAR) rather than attempt to discern a complex interaction from an unusable image. The motivation to enhance robustness for this specific task feels assumed rather than proven.
3. The labels under severe corruption may be not valid. This may be a part missing in the benchmark's design. The paper seems not provide evidence to confirm that the ground-truth HOI triplets are still perceptible or valid after severe corruption is applied. Looking at the examples (e.g., fig.3), it is questionable whether a human annotator could still identify the original interaction. So, is the model's performance dropping because it is not robust, or because the task has become impossible on some samples?
4.  The novelty of the CRI metric. The paper claims the Composite Robustness Index is a novel contribution. However, its normalizing the mean corrupted performance by the clean-set performance, seems frequently seen in previous works [a, b] (often called "effective robustness" or "relative accuracy"). While adding a variance penalty is a reasonable tweak, claiming the metric as a novel contribution may be an overstatement. The paper also fails to compare CRI against other existing, similar metrics.
5.  The contributions seem entangled. The paper introduces both a benchmark (RoHOI) and a training method (SAMPL) to solve it. This may cause a problematic entanglement: for example, the benchmark's "Occlusion" and "Screen Crack" corruptions are precisely the type of challenge that a semantic masking strategy is designed to solve. This feels like "teaching to the test," and it's unclear if SAMPL provides general robustness or just adaptation to some specific simple corruption types.
6. Unfair baseline comparison. Following the point above, the SOTA comparison in Table 1 is arguably unfair. SAMPL is an augmented training strategy applied on top of the RLIPv2 baseline. The other models (UPT, CDN-L, etc.) are, by contrast, evaluated "out-of-the-box" without a comparable robustness-focused augmentation strategy. A fair comparison would have been to apply other SOTA augmentation methods (e.g., AugMix, AutoAugment) to the baselines or to apply SAMPL to other backbones to show its generalizability.
7. Based on point 6 above, the authors do not compare SAMPL to standard robustness techniques (e.g., adversarial training, AugMix), making it difficult to assess its relative contribution.
8. Since the authors claim the robustness issue exists for the task, not just a method, and the proposed method is conceptually not baseline-specific, they may consider verifying the proposed method over different existing HOI methods to show its effectiveness.
9. In the benchmark, some corruption types are too simple and far from the real-world scenarios. For example, the occlusion setting is constructed purely by adding some black blocks over the image.

[a] The Evolution of Out-of-Distribution Robustness Throughout Fine-Tuning

[b] COCO-O: A Benchmark for Object Detectors under Natural Distribution Shifts

**Questions:**

To address the concerns raised above, I would like to mention that the questions below may be noteworthy:

1.  Have you conducted any quality control or human-subject studies to measure the perceptibility or ambiguity of the ground-truth HOI labels after applying Level 4 and Level 5 corruptions? How can you guarantee that the performance drop you are measuring is due to model failure and not data/label invalidation?
2.  Can you provide a more concrete and compelling use-case from a robotics or autonomous driving context from actual application scenarios, where detecting a specific HOI (and not just the objects) from a severely corrupted image is important and existing autonomous driving/robot solution would be an inadequate solution?
3. How does CRI substantially differ from, or improve upon, existing "effective robustness" metrics in the literature that also normalize by clean-set performance?
4.  How can you disentangle the SAMPL method's contribution from the benchmark's design? What is SAMPL's performance (and the baselines') on various, actual robustness settings? This would be a much stronger test of its "general" robustness.
5. The authors does not provide the dataset (or a subset of it) for review.

---

### Official Review · Reviewer_W6pD · 2025-11-06

**Soundness:** 2
**Presentation:** 2
**Contribution:** 2
**Rating:** 4
**Confidence:** 3

**Summary:**

This paper studies Human-Object Interaction under visual corruptions. The author first built a benchmark containing 20 corruption types by manipulating images in HICO and V-COCO. To benchmark the robustness of HOI detection models, the authors introduced two metrics. Experiments reveal that most HOI detection models suffer from visual corruptions. To improve the models' robustness against visual corruptions, the authors proposed a progressive learning training strategy, SAMPL. Experiments showed that SAMPL successively improve RLIPv2's performance on both high-quality and low-quality images. Ablation studies justified the detailed training recipe.

**Strengths:**

1. The authors proposed benchmarking HOI detection models' robustness against visual corruptions, which is practical and useful for real-world robots.
2. The benchmark is comprehensive, covering many scenarios.

**Weaknesses:**

1. The authors proposed a progressive training strategy, but did not compare it against some straightforward recipes, *e.g.*, directly adding corrupted images into the training set and training from scratch. The lack of such straightforward baselines impairs the necessity of the proposed method.
2. The corruption types seem not natural enough. See Questions.
3. Some of the claims in Sec. 5 are superficial. See Questions.
4. The experiment is questionable. See Questions.

**Questions:**

1. How will the model perform if the authors directly add images with different corruption levels into the original dataset and train the model with this large dataset?
2. As I understand, environmentally-induced effects are caused by the environment. However, rainbow effect and vignette effect are more likely to be caused by the deficiency of the imaging system. The blur caused by rain and fog fit the category of environmentally-induced effect better. As for occlusion, there have been strong arguments on the sparsity of the semantics of HOI [1]. Using occlusion as a corruption type can raise great concern. If the key interacting area between a human and an object is blocked as in Fig.1, what should be the GT? On the contrary, there is another type of occlusion: a person sitting at the dining table with his body and legs occluded by a big chair and only his head present in the image. I personally think such kind of occlusion is the least controversial.
3. How do the authors interpret the role vision-language pretraining plays in HOI detection? As we know, CLIP is trained on curated 400M image-text pairs, which form quite a large dataset. If the authors really want to claim that vision-language pretraining is the key, they should prove that models trained with the same amount of language-free data can not achieve comparable performance. As previous models have different architectures and are trained with different datasets from RLIPv2, a lack of detailed ablation study can not support the claim. Please refer to [2].
4. What is the performance of the proposed method on some larger or better-performing models, such as RLIPv2 Swin-L? As there are more and more models with more parameters or better performance, researchers and developers may directly choose these models. If the proposed method can not boost the performance on larger foundation models, the impact of the method is still limited.
5. The role of architecture scaling is still questionable. Can the authors rule out the influence from data used to train UPT and GEN-VLKT?

Reference:

[1] Li, Y.L., Liu, X., Wu, X., Li, Y., Qiu, Z., Xu, L., Xu, Y., Fang, H.S. and Lu, C., 2022. HAKE: A knowledge engine foundation for human activity understanding. IEEE Transactions on Pattern Analysis and Machine Intelligence, 45(7), pp.8494-8506.

[2] Fan, D., Tong, S., Zhu, J., Sinha, K., Liu, Z., Chen, X., Rabbat, M., Ballas, N., LeCun, Y., Bar, A. and Xie, S., 2025. Scaling language-free visual representation learning. In ICCV, 2025.

---

### Official Review · Reviewer_S6mF · 2025-11-07

**Soundness:** 3
**Presentation:** 3
**Contribution:** 1
**Rating:** 2
**Confidence:** 5

**Summary:**

This paper presents RoHOI, a robustness benchmark for human-object interaction detection. Since HOI models are usually trained and evaluated on a very intense-labeled clensed dataset (usually V-COCO and HICO-DET), its robustness against various occlusions and unexpected noise are unexplored. This paper presents a unique benchmark that is used to evaluate HOI detection under 20 corruption types which generate challenging environments for robust scene understanding. Also, it proposes a learning strategy (SAMPL) to guide the model to be optimized based on holistic and partial cues, following many other vision partial view learning in order to improve HOI detection performance.

**Strengths:**

I think this paper is a interesting step towards the improvement in HOI detection.

Overall, the paper is well written and well organized, and the key ideas are well supported with experiments and details within the paper.

**Weaknesses:**

The most biggest concern I have regarding this paper is that the semantic level corruptions (motion blur, glass blur, sensor, compression, ,transmission distortion, etc.) are actually artifacts that are not deeply bonded to visual relationships, but rather an artifact that disturbs the recognition for individual objects.

A primary concern with this paper lies in its approach to semantic-level corruptions. The types of perturbations considered - such as motion blur, glass blur, sensor noise, compression artifacts, and transmission distortions - are largely low-level visual artifacts that interfere primarily with the appearance and recognition of individual objects. While these corruptions undoubtedly degrade overall visual quality, they are unable to target the object independent "relational" semantics that underpin human–object interaction (HOI) understanding. In other words, these perturbations tend to hinder object detection rather than truly challenge the model’s ability to reason about interactions between entities.

For a robustness benchmark in HOI detection to be genuinely meaningful, the corruptions should focus on interaction-specific disturbances - that is, those that selectively impair relational reasoning while preserving object recognizability. Simply cluttering or degrading the entire image may reduce detection accuracy, but it does so by confounding object appearance rather than testing the model’s capability to infer or maintain visual relationships. Thus, a more insightful direction would be to design perturbations that maintain the integrity of object features yet disrupt contextual or spatial cues essential for understanding interactions [1,2,3].

From this perspective, robustness evaluation in HOI detection should evolve toward interaction-aware corruption design, reflecting the dual dependence of HOI models on both object perception and contextual understanding [4,5]. The most compelling challenge would be to identify or synthesize artifacts that do not compromise object recognition but do impair relational inference - for instance, by altering spatial configurations, human pose cues, or affordance-related context. Such an approach would better capture the unique nature of HOI robustness and push models toward deeper relational understanding beyond low-level perception.

Regarding the proposed SAMPL method, it appears to operate primarily as a scheduled learning strategy that leverages object-level masking and partial-view optimization. While this approach contributes to robustness at the level of object visibility, it aligns more closely with efforts in robust visual relationship learning under conditions of incomplete object perception rather than addressing the broader challenge of relational degradation. A more ambitious extension would be to develop mechanisms that explicitly disentangle and preserve interaction representations even when contextual or relational cues are perturbed—thus achieving robustness not merely to visual corruption but to semantic disruption.

[1] Benchmarking Adversarial Robustness on Image Classification, CVPR'20

[2] Shuffle-Then-Assemble: Learning Object-Agnostic Visual Relationship Features ,ECCV'18

[3] Visual Relationship Transformation, ECCV'24

[4] Consistency Learning via Decoding Path Augmentation for Transformers in Human Object Interaction Detection, CVPR'22

[5] DDS: Decoupled Dynamic Scene-Graph Generation Network

**Questions:**

Q1. Could there be a design that could selectively corrupt the relationships in an adversarial manner? Currently the corruptions are pre-categorized, but could there be a way that we could selectively assemble the corruptions [1,2] that could preserve object features while selectively disturbing visual relationship recognitions?

[1] Understanding Robustness of Transformers for Image Classification, ICCV'21

[2] Measuring Robustness to Natural Distribution Shifts in Image Classification, NeurIPS'20

---

### Note · Authors · 2025-11-12

**Comment:**

Dear Area Chair and Reviewers,

Thank you for overseeing our submission and for the thoughtful, constructive feedback. Based on the reviews, we will undertake a substantial revision, in particular refining interaction-aware corruption design. Therefore, we will withdraw Submission 10705 and prepare a thoroughly revised version for a future cycle. We are grateful for your time and insights, and we wish you a smooth remainder of the review cycle.

Best regards,

Authors of Submission 10705

**Withdrawal Confirmation:**

I have read and agree with the venue's withdrawal policy on behalf of myself and my co-authors.